# Types of Gastric Carcinomas

**DOI:** 10.3390/ijms19124109

**Published:** 2018-12-18

**Authors:** Helge L. Waldum, Reidar Fossmark

**Affiliations:** 1Department of Clinical and Molecular Medicine, Faculty of Medicine and Health Sciences, Norwegian University of Science and Technology, 7006 Trondheim, Norway; reidar.fossmark@ntnu.no; 2Department of Gastroenterology and Hepatology, St. Olav′s University Hospital, 7006 Trondheim, Norway

**Keywords:** classification, gastric carcinoma, gastrin, neuroendocrine cells, types of gastric carcinomas

## Abstract

Gastric cancer has reduced prevalence, but poor prognoses. To improve treatment, better knowledge of carcinogenesis and cells of origin should be sought. Stomach cancers are typically localized to one of the three mucosae; cardial, oxyntic and antral. Moreover, not only the stem cell, but the ECL cell may proliferate and give rise to tumours. According to Laurén, the classification of gastric carcinomas seems to reflect biological important differences and possible different cell of origin since the two subtypes, intestinal and diffuse, do not transform into the other and show different epidemiology. The stem cell probably gives rise to the intestinal type, whereas the ECL cell may be important in the diffuse type. Elevation of gastrin may be the carcinogenic factor for *Helicobacter pylori* as well as the recently described increased risk of gastric cancer due to proton pump inhibitor treatment. Therefore, it is essential to determine the role of the gastrin target cell, the ECL cell, in gastric carcinogenesis. Clinical trials with gastrin antagonists could improve prognoses in those with gastrin receptor positive tumours. However, further studies on gastric carcinomas applying relative available methods and with the highest sensitivity are warranted to improve our knowledge of gastric carcinogenesis.

The mucosa in the gastric fundus and corpus (oxyntic mucosa) is very different from that of the antrum. Whether the cardiac mucosa constitutes a third type of mucosa, or just represents metaplasia, has been disputed [1]. Nevertheless, there are at least two different mucosae in the stomach (oxyntic and antral), that makes it peculiar that carcinomas originating in the stomach in most contexts have been lumped together as gastric carcinomas. In general, there has been little effort to make a distinction between carcinomas originating in the antral and oxyntic mucosae. However, the border between oxyntic mucosa and antral mucosa is not so sharp as previously thought [2]. In general, if any distinction between gastric carcinomas based upon localization has been made, this has been between proximal tumours localized to the cardiac region and distal tumours including those in the oxyntic and antral mucosae.

Furthermore, many different cell types in the gastric mucosa have the ability to divide and thus give rise to carcinomas. Besides stem cells found in both the oxyntic and antral mucosae, neuroendocrine (NE) cells spread between the other epithelial cells can also divide. The ability of NE cells to divide has been most convincingly shown for the histamine producing enterochromaffin-like (ECL) cell [3], which is one of the most abundant NE cells in the stomach. This ability has also been indirectly indicated by the case reports describing ghrelinomas (developing from A-like cells) [4]. Ideally, tumours have to be classified not only according to organ of occurrence, but also mucosa type and cell of origin. In the present review, we will discuss classification of gastric carcinomas based upon that perspective.

## 1. Present Classifications

There are many different classification systems for gastric carcinomas. Classification may be based upon gross appearance (polypoid, fungating, ulcerated and infiltrative) as described by Borrmann [5]. This old classification is still useful, and the macroscopic growth pattern has some relevance to the microscopic classification systems. Gastric carcinomas show microscopically resemblance to intestinal mucosa; a feature that is included in most of the microscopically based classification systems. The World Health Organization (WHO) classification system distinguishes between papillary, tubular, mucinous and signet-ring cell subtypes [6]. This classification is used by many and may be useful. In the 1960s, Laurén made a classification of gastric carcinomas according to the presence of glandular growth pattern (intestinal type) and the lack of such growth (diffuse type) [7]. Both types were, however, believed to develop from an exocrine cell linage since they both were histochemically PAS positive, a hitherto accepted exocrine marker. Subsequently both types were classified as adenocarcinomas. There is a problem with Laurén′s classification in that 15–20% of the carcinomas cannot be classified into either group. Finally, Ming classified gastric carcinomas into two types, those with an expanding and those with infiltrative growth pattern [8]. However, Laurén′s classification seems to reflect biological differences as they during growth do not transform into the other [7], and epidemiologically show different trends [9]. The World Health Organization (WHO) classification is in addition to Laurén′s classification most often used.

During the last decades, improvement in molecular technology has made it possible to determine mutations in carcinomas and thus make classification upon these findings. Such a classification was published in a large and impressive study a few years ago [10]. Mutation analyses of tumours may guide treatment strategy selection, particularly when driver mutations are found [10]. Mutation analyses do not necessarily give indication of the cell of origin [11]. It is conceivable that similar mutations may be central in the carcinogenesis in general and occur in different cell types during tumourigenesis. Therefore, both mutation analyses of malignant tumours as well as analyses of actual markers with respect to cell of origin and its growth regulation, will be useful both with respect to treatment, prophylaxis and prevention of the carcinomas.

## 2. Cells in the Oxyntic Mucosa with the Ability to Divide and Their Growth Regulation

The stem cell in oxyntic mucosa is localized to the isthmus region of the glands and gives rise to all types of exocrine cells in the oxyntic mucosa. The origin of the endocrine cells in the oxyntic mucosa is disputed [12]. This is in contrast to the antral mucosa where the stem cell—experimentally—also has been shown to develop into endocrine cells [12]. The mature cells differentiated from the stem cell principally do not divide although there is some evidence for proliferation of some chief cells [13] playing a role in the development of the so-called spasmolytic polypeptide-expressing metaplasia (SPEM). However, there is also a study casting doubt upon the role of chief cells in proliferation and tumourigenesis [14]. Among the endocrine cells, the ECL cell has been clearly shown to proliferate [3], which probably is a general ability of NE cells.

## 3. Cells in the Antral Mucosa with the Ability to Divide

Stem cells localized to the gland in the antral mucosa probably develop into all epithelial cell types in the glands including NE cells [15], which consist of G-, D- and A-like cells. 

## 4. Etiology of Gastric Carcinomas

As noticed a long time ago, gastric carcinomas seldom develop in a stomach without gastritis [16]. With the identification of *Helicobacter pylori* (Hp) as the main cause of gastritis [17], it was to be expected that Hp soon was recognized as the most important factor in gastric carcinogenesis [18] both for carcinomas of diffuse and intestinal types [19]. However, Hp infection confined to the antral mucosa causes duodenal ulcer, but apparently protects against development of gastric carcinoma [20]. Therefore, it seems that Hp infection is not carcinogenic per se [21]. Moreover, it has become clear that oxyntic atrophy is essential for gastric cancer [22] including the carcinogenic effect of Hp infection [23]. Oxyntic atrophic gastritis reduces gastric acid secretion, and therefore indicating that Hp′s carcinogenic effect most probably is related to gastric hypoacidity. Since the main function of gastric acid is to kill swallowed microorganisms [24], it could be that secondary infections could play a role in gastric carcinogenesis. If so, it would be expected that Hp infection with atrophic oxyntic gastritis would predispose to carcinoma irrespective of anatomic localization and type of mucosa. However, Hp infection does not predispose to gastric carcinoma localized to the cardiac region [25]. Furthermore, so-called autoimmune gastritis which is confined to the oxyntic mucosa where it causes total glandular atrophy, predisposes to gastric carcinomas only in the oxyntic mucosa [26]. Accordingly, secondary microbial infection cannot explain the carcinogenic effect of oxyntic atrophic gastritis either due autoimmune or Hp induced oxyntic atrophy. On the other hand, oxyntic atrophy reduces gastric acidity leading to hypoacidity and hypergastrinemia. The target of gastrin is the oxyntic mucosa and more precisely the ECL cell [27]. Gastrin regulates ECL cell function, histamine release [28] and concomitantly, the ECL cell proliferation [29]. Long-term hypergastrinemia causes ECL cell hyperplasia in rodents [30,31] as well as in humans [32], which in time leads to ECL cell-derived tumours and gastric NE cell tumours (NETs) in all examined species [30,31,33]. In short-lived species like rodents, long-term hypergastrinemia caused tumours that initially were classified as carcinomas [34]. These tumours were reclassified as NETs when the mechanism and cell of origin were recognized [31]. Quite recently, the role of long-term treatment with potent inhibitors of gastric acid secretion, the proton pump inhibitors (PPIs), were reported to predispose to gastric carcinoma also in humans [35,36]. Previously, only a case report had described gastric carcinoma in PPI users [37]. Since the biological effect of PPIs is to inhibit gastric acid secretion and thus reduce gastric acidity, it is reason to assume that the gastric carcinogenic effect of PPIs also is mediated by hypergastrinemia. In fact, in a family where both parents were heterozygote for a missense mutation in the proton pump alpha subunit, homozygote offspring for this mutation were hypergastrinemic and developed gastric tumours from the age of 23 years [38,39]. Thus, gastrin seems to be involved in most cases of gastric carcinomas occurring anally/distally to the cardiac region, where Hp seems not to play any role [25].

The way gastrin stimulates acid secretion was disputed for many decades. Now, it seems decided that gastrin works by stimulating release of histamine from the ECL cell [28,29] where the gastrin (CCKB) receptor is localized [40]. The parallel positive functional and trophic effects induce hyperplasia and tumours of different degree of malignancy originating from the ECL cell. However, there is no doubt that gastrin affects the oxyntic mucosa in general and not only the ECL cell. The general trophic effect of gastrin on the oxyntic mucosa was indirectly shown by the large, succulent folds found in patients with Zollinger–Ellison syndrome [41] and experimentally in animal studies [29,42]. Whether the trophic effect on the oxyntic mucosa represents a direct effect of gastrin on the stem cell [43] or is an indirect one mediated by a release of signal substances from the ECL cell, is not yet solved. REG I protein released from the ECL cell upon gastrin stimulation is an actual candidate for being the intermediary [44]. However, there is no indication of a gastrin receptor on the parietal cell, neither functionally [45] nor trophically [46].

Gastrin therefore may be the common pathogenic factor for gastric carcinomas originating from the oxyntic mucosa, where there are cells that are stimulated by gastrin receptor agonist. In the antrum, however, there may be a gastrin receptor on the G cell, which presumably is inhibitory as G cell density is increased in gastrin-cholecystokinin knockout animals [47]. It is accordingly difficult to conceive that gastrin should predispose to carcinoma developing from the antral mucosa. Nevertheless, the boundary between oxyntic and antral mucosa is not sharp with oxyntic glands also detected in antral mucosa [2].

Not only Hp, but also other microorganisms like the Epstein–Barr virus may predispose to gastric cancer [48]. Being the first site exposed to food over time, it is natural that diet habits have been studied and incriminated in gastric carcinogenesis. However, taking into consideration the significant role of microorganisms in gastric carcinogenesis, it is difficult to distinguish between carcinogenic factors in the food and microbial contamination.

## 5. Pathogenesis of Gastric Carcinomas 

Gastrin is central in gastric carcinogenesis, and being a peptide hormone, it can directly only affect cells with a gastrin receptor. Gastrin stimulates histamine release from, and proliferation of, the ECL cell. Every condition with long-term hypergastrinemia in animals as well as humans predisposes to gastric malignancy [49]. A direct carcinogenic effect of Hp on the gastric mucosa may be excluded since infection only in the antrum on the contrary, protects against gastric cancer [20]. Moreover, even with inflammation in the oxyntic mucosa, a predisposition to gastric cancer requires that atrophic gastritis has developed. If gastric hypoacidity should predispose to gastric cancer by secondary microbiological infections, it would be expected that the tumours should develop in the whole stomach, and not only in the oxyntic mucosa [26]. Accordingly, hypergastrinemia itself could be the pathogenic factor for carcinoma development secondary to Hp infection [21,49]. If so, the sole established target cell for gastrin, the ECL cell, must play an important role in gastric carcinogenesis. This has been a question of peculiar controversy, and it was even claimed that the ECL cell in man did not proliferate [50] although it was clearly shown that this cell proliferates in rodents [3]. Since the ECL cells occur in clusters in hypergastrinemic patients, it was difficult to understand the reluctance to accept that the ECL cell does divide. It has subsequently been accepted that the ECL cell proliferates and gives rise to gastric NETs. However, the role of ECL cells in gastric carcinogenesis in general has been denied except for NE carcinomas (NECs). Nevertheless, gastric NETs and gastric carcinomas occur both with increased frequency in patients with autoimmune gastritis [26,33], and when applying immunohistochemistry with improved sensitivity [51] we could show that gastric carcinomas in patients with pernicious anemia expressed NE markers [52]. In a larger study on gastric carcinomas in general, we could conclude that an important proportion actually could be classified as NE tumours, and then particularly those classified as diffuse according to Laurén [53]. Among the gastric carcinomas of diffuse type, NE differentiation is particularly pregnant in the signet ring subgroup [54]. These carcinomas are PAS positive, but do not express specific markers for mucin [55]. It is remarkable that experts seem to rely more on unspecific histochemical methods than more specific one like immunohistochemistry and in-situ hybridization, in classification of tumours [56]. Based upon our results it seems that the major part of gastric carcinomas of diffuse type may develop from the ECL cells secondary to long-term hyper-stimulation with gastrin. Although some gastric carcinomas of intestinal type also express NE markers [53], most of them do not. Nevertheless, gastrin and the ECL cell could be involved in the tumourigenesis of gastric carcinomas of intestinal type by gastrin releasing not only histamine, but also a factor like REG I [44] which has a stimulatory effect on stem cell proliferation and thus could predispose to carcinoma development [57]. Moreover, as stated before, children born with a missense homozygote mutation in one of genes coding for the proton pump and thus being anacidic and hypergastrinemic from birth, developed ECL cell NETs and gastric carcinoma at young age [38,39] further substantiating the central role of gastrin and the ECL cell in gastric carcinogenesis. Similarly, the recent epidemiological studies describing that patients having used PPIs for long-term have increased risk for gastric cancer [35,36] also supports such a view. A combination of Hp infection and PPI treatment results in an additive increase, at least in serum gastrin [58]. The gastrin receptor is expressed on tumor cells both in diffuse and intestinal type of cancer [59], which indirectly support the presence of a gastrin receptor also on the oxyntic stem cell. However, very recently it was reported that the enterochromaffin (EC) cell in the small intestine, a cell closely similar to the ECL cell, participated in stem cell dynamics [60] indicating a possible direct role of the ECL cell also in intestinal type of gastric cancer. Interestingly, Goetze and co-workers showed that a majority of gastric carcinomas localized to different parts of the stomach, expressed both gastrin and the gastrin receptor [61] that could indicate a stimulatory autocrine loop. In the same line, Hayakawa and co-workers described that antral stem cells expressed the gastrin receptor that could be stimulated by progastrins but not by amidated gastrins, and that this stimulation played a role in carcinogenesis [62]. However, it is difficult to conceive that the same gastrin receptor at some locations should have affinity for amidated gastrin and not to progastrins and at other cellular locations bind progastrin and not amidated gastrins. From a biological point of view, a gastrin receptor on antral cells (G cells) would be expected to be inhibitory and not stimulatory. In any way, before a role of autocrine stimulation and progastrins in antral carcinogenesis can be accepted, it needs to be shown in vivo.

Besides gastrin, also genetic factors do of course play an important role also in gastric carcinogenesis. This is best exemplified for mutations in E-cadherin gene (CDH 1) which when homozygote, result in gastric carcinomas of diffuse type at an early age [63]. Based upon the fact the ECL cells normally occur single among other epithelial cells, we examined human ECL cells for E-cadherin expression, which we could not detect [64]. This property, together with paracrine effects of substances released like histamine, could predispose this cell to invasiveness and metastasis, central factors in gastric carcinomas of diffuse type. Moreover, there is no accepted marker for stem cells in the oxyntic mucosa [65]. Accordingly, there is no experimental support for the common belief that oxyntic gastric carcinomas are of stem cell origin.

A minor part of gastric carcinomas may be induced by EBV [48]. EBV, being a DNA virus, may induce tumour growth by incorporating virus DNA fragments into the genome of cells having the ability to divide. A direct role in carcinogenesis is well known for viruses in contrast to bacteria, which have hitherto not been shown to play such a role.

## 6. Intestinal Type of Gastric Carcinoma

*Helicobacter pylori* infection plays a central role in the etiology of intestinal type of carcinomas [66]. Typically, the intestinal subtype develops on a background of atrophic gastritis often with intestinal metaplasia. There is some uncertainty concerning the role of metaplasia in gastric carcinogenesis. Thus, incomplete gastric intestinal metaplasia was shown to imply a greater risk than complete metaplasia [67]. Intestinal metaplasia may be a direct precursor of cancer or alternatively only be a marker of long-term gastric atrophy. Interestingly, the fall in incidence in gastric carcinoma seen during the last decades mainly reflects a fall in occurrence of intestinal type of cancer [68]. The reduction in Hp infection in most of the developed countries is probably the principal factor in this decline in incidence. However, it may be that the decline started before the reduction in Hp infection in the population [69]. It is, however, probable that the diagnosis gastric carcinoma was based upon less precise diagnostic criteria before the 1960s, thereby including other malignant intra-abdominal carcinomas among gastric cancers. The role of Hp in gastric carcinogenesis, although accepted the most important one, may even presently be underestimated since no method used to diagnose Hp has hundred percent sensitivity. Moreover, when atrophic gastritis has reached a stadium where virtually all glands are destroyed, gastric anacidity develops; a condition under which Hp cannot live [70]. Therefore, to estimate the real role of Hp infection in the carcinogenesis of carcinomas of intestinal type, a combination of the most sensitive tests should be applied, and gastric acidity should be determined or assessed indirectly by careful histological examination of the mucosa outside the tumour. The way Hp induced oxyntic atrophy with secondary hypoacidity and hypergastrinemia could induce gastric carcinomas of intestinal type is depicted in Figure 1. 

Gastric carcinomas occur more frequently in men compared to females, but this difference is less pronounced now since the intestinal type, which is more common in men, is the type with a falling incidence [68]. The reason for sex difference in the incidence of gastric carcinoma of intestinal type is unknown. On the other hand, gastric carcinomas of diffuse type are not uncommon in young women [68].

## 7. Diffuse Type of Gastric Carcinoma

Gastric carcinoma of diffuse type typically invades the submucosa at an early phase and the tumour cells often spread in the upper layers of the stomach wall instead of growing as a tumour protruding into the lumen. Therefore, the diffuse gastric carcinomas may be easily overseen at early phases even at endoscopy. The tumour cells induce fibrosis, which reduce elasticity and compliance leading to the well-known typical symptom of gastric carcinoma of scirrhous cancer, early satiety. Thus, diffuse gastric carcinomas differ from the intestinal ones, not only by lack of glands but also by a marked fibrosis. The cancer cells accordingly must release a factor stimulating fibrosis, indicating different cells of origin for the two types of gastric cancer. Based upon a common expression of neuroendocrine markers in the carcinoma cells of diffuse gastric cancer [51,52,53,71,72], and that the PAS positive material in the signet ring cell carcinomas, a subgroup of carcinomas of diffuse type, is not mucin [55], we have argued that the gastric carcinomas of diffuse type actually are NE tumours [49,53,56]. The most prevalent NE cell in the oxyntic mucosa is the ECL cell, and oxyntic gland atrophy either caused by Hp infection or autoimmune gastritis, predisposes both to more benign gastric NETs (ECL derived) [33,73] and gastric carcinomas [18,26]. Interestingly, the ECL cell produces basic fibroblast growth factor (bFGF) [74]. The expression of bFGF seems to be particularly strong in gastrin-stimulated ECL cells as found in ECL cell hyperplasia and gastric neuroendocrine tumours (NETs) [74]. In addition, gastric carcinomas, and particularly those of the diffuse type subgroup scirrhous, express bFGF [75,76]. In general, gastric carcinomas of diffuse type are more prone to express bFGF [76]. In a case report, Huyodo et al. [77] speculated that the aggressive fibrosis seen in scirrhous cancer could be due to bFGF.

The adhesion molecule E-cadherin is expressed in epithelial cells where it contributes to the continuity of epithelial surfaces. Hereditary diffuse gastric cancer of diffuse type is caused by a missense mutation in the gene, CDH1, coding for E-cadherin [63]. The neuroendocrine (NE) cells in the different mucosae are found spread among the other epithelial cells suggesting reduced adherence to each other compared with the other epithelial cells. Therefore, we examined E-cadherin expression in dispersed oxyntic mucosal cells and found that neuroendocrine cells identified by immunocytochemistry using chromogranin A antibodies, did not express membranous E-cadherin [64]. Thus, normal NE cells may be predisposed to both invasion and metastasis malignancy. E-cadherin mutations were described in 50% of gastric carcinomas of diffuse type, but not in tumours of intestinal type [78]. Abnormal expression of E-cadherin occurs even in early gastric carcinomas [79], and abnormal E-cadherin immunoreactivity occurs markedly more frequently in tumour cells of diffuse type compared with those of intestinal type [80]. Moreover, sporadic gastric carcinomas of diffuse type occurring in patients below the age of 45 years, so-called early onset gastric carcinomas, showed alteration in CDH1 gene [81]. 

The cellular identification of the gastrin receptor has been difficult due to nonspecificity of the antibodies used. Based upon functional and trophic studies [27,46] as well as infusing a labelled gastrin analogue in the isolated rat stomach followed by immunohistochemistry, we could show that the gastrin receptor was localized on the ECL cell and not the parietal cell in the rat [40]. However, the presence of a gastrin receptor on the endodermal stem cell localized to the neck of the glands, could have been missed by this study due to the scarcity of such cells. We detected gastrin receptor expression both by immunohistochemistry and in–situ hybridization both in diffuse and intestinal type of gastric carcinomas [59], which may indirectly suggest that there is a gastrin receptor on the stem cell. Anyhow, gastrin and the ECL cell seems to be central in the tumourigenesis of diffuse gastric carcinomas (Figure 2). 

The differences between gastric carcinomas of intestinal and diffuse types of gastric cancer are summarized in Table 1.

## 8. Conclusions 

The present classification systems of gastric carcinomas do not imply any appreciable differences in prognoses or choice of therapy. Moreover, the prognoses in gastric carcinoma has remained frustratingly poor. Therefore, it is evident that major improvements are needed. We recommend that at surgery, the localization of the tumour should be classified as cardial, oxyntic or antral although a correct anatomical and mucosal classification may not always be possible since the border between oxyntic and antral mucosae is not sharp. Furthermore, we mean that tumours should be histologically classified according to Laurén as it seems to reflect stable biological differences. All tumours ought to be examined for endocrine differentiation by immunohistochemistry with antibodies towards general NE markers like chromogranin A and synaptophysin. NSE is also a sensitive marker, and its specificity is better than generally believed [83]. Finally, the detection of gastrin receptor by immunohistochemistry and especially by in-situ hybridization, will also demonstrate a possible role of gastrin in the carcinogenesis and a possible treatment strategy. Based upon the presented data it seems that the diffuse and intestinal types of gastric cancer are two different tumours, although overlaps exist. However, further meta-analyses and prospective studies are warranted, aiming at improving the diagnostic sensitivity by integrating high sensitivity criteria and parameters from different classification methods, including microscopic morphological patterns and private DNA mutations.

## Figures and Tables

**Figure 1 ijms-19-04109-f001:**
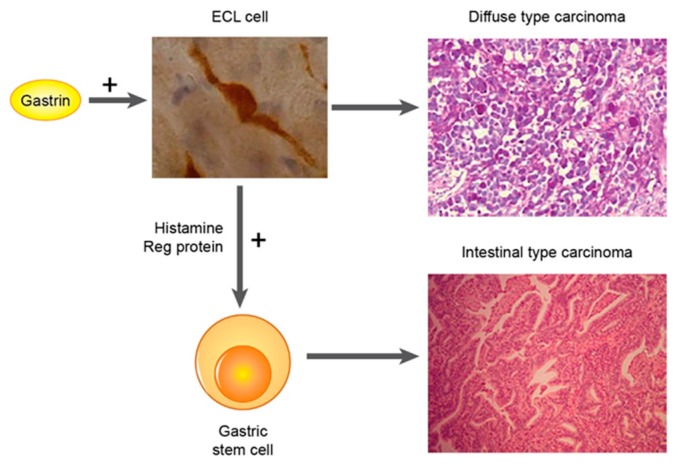
Proposed carcinogenesis for gastric carcinoma of intestinal type (From ref 19 with permission [21]).

**Figure 2 ijms-19-04109-f002:**
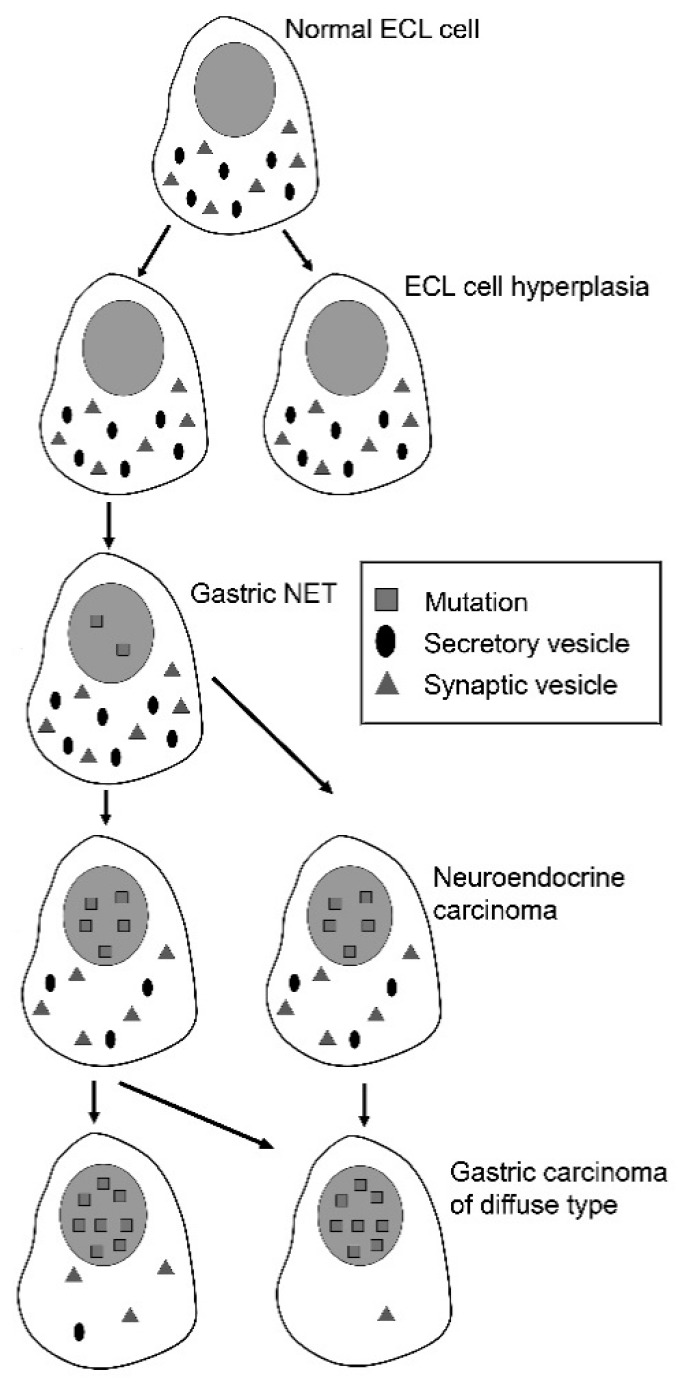
Proposed carcinogenesis for gastric carcinomas of diffuse type (From ref 78 with permission [82]).

**Table 1 ijms-19-04109-t001:** Types of gastric carcinomas.

	Intestinal Type	Diffuse Type
**Glands**	Yes	No
**Age**	Older	Young
**Transition from one to the other**	No	No
**Macroscopic growth pattern**	Tumour into the lumen	Tumour spreads along the mucosa
**Fibrosis**	Not marked	Marked
**Endocrine markers**	Seldom	Often
**bFGF**	Seldom	Often
**E-cadherin present**	Often	Seldom
**Cell of origin**	Stem cell?	ECL cell?

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
