# Peer review of "Types of Gastric Carcinomas"

_ijms, 2018, doi:10.3390/ijms19124109_

Round 1
Reviewer 1 Report
The review of Waldum and Fossmark is well written and provides an interesting up-to-date overview and discussion of the different types of carcinomas in the stomach and their pathogenesis. The authors emphasize in this context the role of neuroendocrine elements of the carcinogenesis, not least that of the antral hormone, gastrin. Adding to the quality of the review is the well-known fact that the present authors have systematically studied gastric carcinogenesis for decades and published several important contributions to the understanding of the development of these carcinomas.
A single aspect about carcinomas in the stomach and gastrin is not discussed: That is the local expression of gastrin and its receptor in malignant cells within the cancer tissue. Such expression may very well play an autocrine role in the growth of the neoplasm (see for instance Goetze et al. “Characterization of gastrins and their receptor in solid human gastric adenocarcinomas”. Scand. J. Gastroenterol. 2013;48:688-695). It is desirable that the authors also mention this aspect.
Author Response
Referee I.
We have added a paragraph on autocrine stimulation and added two references connected to this (including the one suggested). Lines 209-218.
Reviewer 2 Report
The authors satisfactorily addressed only some of the points of criticims that I raised in my previous review of their manuscript, while giving little or no attention to a couple of others:
Point 1 of previous review:
The authors disregarded this point, still I deem it indicated that they provide a couple of references in support of their statement;
Point 2 of previous review :<The paper lacks a conclusive section or paragraph, with clearcut recommendation(s). The authors should reformulate lines 272 and following in a sharper, more conclusive way. ... it is my opinion that the authors should adopt a prudent attitude in summarizing current evidence and experience and formulating their recommendations. While legitimately emphasizing their point of view, that the source cell in gastric cancers should be determined by the use of such markers as E-cadherin in epithelial cell-derived cancers and chromogranin A, synaptophysin and neural-specific enolase in cancers of neuroendocrine origin, the authors should show a permissive attitude towards further prospective studies and meta-analyses, aiming at integrating into such a classification system what might be persuasively extracted from other systems, such as WHO’s, both in terms of:
a) microscopic morphology, as the morphological distinction between intestinal and diffuse gastric carcinomas appears to be poor and unreliable;
b) DNA- and RNA-based molecular fingerprinting, with the determination of private mutations.>
The authors addressed this matter of concern in a limited way. I still suggest that they rephrase and expand the last sentence of the conclusions as follows:
" However, further meta-analyses and prospective studies are warranted, aiming at improving the diagnostic sensitivity by integrating high sensitivity criteria and parameters from different classification methods, including microscopic morphological patterns and private DNA mutations."
Point 3 of previous review is related to point 2: <- the="" abstract="" should="" be="" rewritten="" accordingly="">. What precedes (point 2) should be reflected in the abstract.
Author Response
Referee II
1.We have omitted the sentence of Laurén classification seems to be gradually accepted.
2.We have changed the final part in line with the suggestions and particularly adopted his proposal of a final sentence. Lines 341-344
3.Similar changes in the abstract that we have reformulated. Lines 19-21 and 23-24